# Exploring the Lived Experiences of Vulnerable Females from a Low-Resource Setting during the COVID-19 Pandemic

**DOI:** 10.3390/ijerph20227040

**Published:** 2023-11-09

**Authors:** Firoza Haffejee, Rivesh Maharajh, Maureen Nokuthula Sibiya

**Affiliations:** 1Department of Basic Medical Sciences, Durban University of Technology, Durban 4000, South Africa; riveshm@dut.ac.za; 2Division of Research, Innovation and Engagement, Mangosuthu University of Technology, Umlazi 4031, South Africa; sibiya.nokuthula@mut.ac.za

**Keywords:** COVID-19, women’s health, mental health, social interaction, lived experiences, stressors

## Abstract

The onset of the Coronavirus disease 2019 (COVID-19) pandemic has affected the mental health and well-being of women in vulnerable settings. Currently, there is limited evidence that explores the wellness of elderly women under the associated restrictions. This study explores the lived experiences of elderly women in a vulnerable community in Durban, South Africa. A face-to-face, in-depth qualitative approach was implemented to interview 12 women aged 50 years and over. Thematic analysis was used to analyse the data. The findings suggest that social interactions, the effect of a high death rate, and financial strain predominantly affect stress and anxiety levels. Despite the women being in receipt of pensions and/or other grants, their supplementary income was reduced. This, together with the additional expenses incurred during the lockdown, resulted in anxiety over finances. The lack of social interaction, with limits on visiting family and other loved ones when they were ill, along with the limit on the number of people attending the funerals of loved ones were also stressful. This study also reports on the resulting coping mechanisms, which included using hobbies such as baking and sewing as a means of self-care. Religious beliefs also relieved stress while home remedies were used as preventative measures during the lockdown restrictions due to COVID-19.

## 1. Introduction

Coronavirus disease 2019 (COVID-19) was declared a worldwide pandemic by the World Health Organization (WHO) in the year 2020 when incidence levels increased drastically [1]. The onset of the COVID-19 pandemic has affected many on a global scale resulting in variable effects on society, particularly the mental health and well-being of women. Overall current studies have shown that women reported higher levels of stress, anxiety, depression, and post-traumatic stress disorder (PSTD), which severely affected their mental health [2,3]. The wellness of individuals is supported by social interactions, support, intimacy, and the ability to engage in society as a whole [4].

South Africa attempted to curb the high incidence rates of COVID-19 by implementing various restrictions using governmental disaster management policies [5]. These restrictions included no social gatherings, the wearing of masks in public, reduced contact when traveling, time-specific curfews, and a period of complete economic closure by non-food and health businesses. The alarming inception of mental health issues has been addressed by the South African College of Applied Psychology [6], which states that approximately one in six South Africans suffer from anxiety, depression, or substance use disorders. In addition, only 27% of those diagnosed with severe mental health illnesses have access to treatment [6]. The social determinants of mental health are directly proportional to poverty, inequality, gender disparity, access to employment, and interpersonal violence [7]. One of the major confounding effects of the national lockdown included coping with job loss, which was shown to be directly proportional to high levels of mental health issues [8]. The government implemented social restrictions to reduce person-to-person contact and decrease transmission rates [9,10]. Coping mechanisms to prevent anxiety and depression during the COVID-19 pandemic included the effective use of electronic communication via social media, supportive networking, limiting exposure to COVID-19 news, and self-care by occupying oneself with a skill, hobby, or physical relaxation [11,12].

There is a need to protect vulnerable social groups, as there are no strategies to address those who were affected psychologically and physically during the COVID-19 pandemic. It has been shown that women experienced different health risks and implications during the pandemic; these determinants include social, economic, and cultural factors [13]. A recent review of the effects on mental health during the COVID-19 pandemic showed that women are more affected in both their workspaces and at home, resulting in increased workloads due to restrictions and quarantine [14]. The psychological risks associated with COVID-19 have largely affected women, and additionally, studies have reported that women from lower-income groups have higher levels of stress and anxiety [2,14]. The effects of social distancing contributed to feelings of loneliness, especially among the elderly in closed-off settings [15]. This makes the elderly more vulnerable to mental health concerns as they are not well-connected in terms of social media interactions. COVID-19 has been shown to affect communities disproportionately, especially those from low-income settings [16]. Since women live longer than men, this places them in a position of economic vulnerability at an older age, with low-income security issues such as a reliance on government pensions [17]. This is particularly true among those in aged communities, who were susceptible to multiple deprivation conditions during the crisis [18]. The risk of mortality due to COVID-19 increased four-fold in those aged 50–65 and ten-fold in those aged 65 and older, which further increased when comorbidities were involved [19]. Currently, there is a need for governmental policies to address such aberrations of mental health and well-being in those affected by the onset of COVID-19, particularly among vulnerable social groups [10,20,21].

The COVID-19 pandemic continues to affect populations around the world; changes in the socioeconomic environment of society have resulted in significant stressors, particularly among women [1,2,21]. Currently, the South African health systems are still recovering from the pandemic, with most resources focused on the monitoring and tracing of COVID-19 [10]. In South Africa, the province of KwaZulu-Natal reported a COVID-19 prevalence of 17.7%, which was the second-highest prevalence rate among the nine provinces in the country [22]. Furthermore, the province of KwaZulu-Natal reported the highest incidence rates during governmental lockdown restrictions [23].

The aim of this study was to explore the lived experiences of vulnerable women over the age of 50 years in a low-resource setting, located in KwaZulu-Natal, South Africa, during the COVID-19 pandemic.

## 2. Materials and Methods

### 2.1. Study Approach and Study Settings

The study employed a descriptive qualitative approach to gain a better insight into the wellness of women in vulnerable settings. This approach allowed the participants to provide a narrative regarding their experiences throughout COVID-19 governmental restrictions during the period from March 2020 to March 2022. This technique permitted researchers to explore the perspective of women in relation to the COVID-19 pandemic among vulnerable communities, in a naturalistic setting. Vulnerable social groups included women who were from low-income settings. The study was conducted in a private room at the DUT-Homeopathy clinic in Kenneth Gardens, which is situated within eThekwini Municipality, KwaZulu-Natal, South Africa. Kenneth Gardens is a large municipal housing estate, offering subsidized housing to families of low socioeconomic status and to those families in which one or more members have a disability. It comprises predominantly black residents whose most crucial concern is access to health care, as residents are reported to travel more than 20 km to receive medical care from a public hospital [24]. The study was conducted among women from July to August 2023, after the lifting of governmental national restrictions due to COVID-19, to ascertain their wellness levels.

### 2.2. Sampling

Purposive snowball sampling was used to recruit female participants over the age of 50 years from Kenneth Gardens, a low socio-economic housing estate in eThekwini Municipality in the province of KwaZulu-Natal, South Africa. This sampling was deemed appropriate for gaining access to participants with specific characteristics. Recruitment was focused on individuals who matched the demographic, who were initially screened via interviews. Upon completing the interviews, researchers proceeded to implement a respondent-driven sampling method to recruit participants of a similar demographic. The sampling continued until data saturation was attained [25]. The researchers first identified possible participants by engaging with members of the community in which the data was collected. The participants were subsequently sampled purposively, using a snowball approach through those who were initially recruited.

### 2.3. Data Collection

An in-depth, semi-structured open-ended questionnaire was administered to collect sociodemographic details, information based on their experiences during lockdown conditions, and the associated burden of COVID-19. Interviews were conducted in either English or isiZulu, audio-recorded, and subsequently transcribed verbatim and translated into English for consistency. Quantitative data relating to demographic and socioeconomic status were collected, focusing on: gender; race; age; number of children/dependents; number of people in household; and source of income. The qualitative approach utilized an open-ended interview conducted with a series of eight questions, probing to determine the outcome variables. The key questions focused on the participants’ lived experiences during the governmental lockdown restrictions (stress-related occurrences, experiences of positive feelings, and the effects of social interactions), financial management (sources of income, care, and treatment), and the various coping mechanisms. These questions pivoted around various open discussions until saturation was achieved. The study implemented Tesch’s eight-step approach for thematic data analysis [26]. The researchers independently familiarized themselves with all the interview transcripts, whereby the synthesis of notes, key points, and relevant statements were highlighted. Descriptive codes were generated from the transcripts to address the objectives of the study. These were validated independently by all co-authors. Thereafter, the coded categories were allocated into defined themes and subthemes. Quality assurance was maintained throughout the dissemination of the study. The study adopted the four-dimensional application criteria of Lincoln and Guba [27], which are important in developing the trustworthiness of qualitative research, namely, the credibility, dependability, confirmability, and transferability of the study. Furthermore, triangulation discussions and immersive involvement among researchers were captured to improve trustworthiness when reporting. This study used the Standards for Reporting Qualitative Research as a guideline to synthesize the structure of the report [28].

### 2.4. Ethical Considerations

All participants of the study were provided with a letter of information, and they provided written informed consent prior to commencing the interview. All participants remained anonymous and were assigned a participant number. Furthermore, recommendations of contacts for social services/clinics were provided to those who required therapy. The principles of autonomy, non-maleficence, beneficence, and justice have been maintained throughout the study, in accordance with the ethical guidelines identified by Polit and Beck [29]. All participants were provided with a grocery hamper to the value of ZAR 50 (South African Rands), to compensate them for their time, after the interview was concluded.

## 3. Results

Twelve female participants were interviewed. All of them were over the age of 50 years. Their full demographic details are provided in Table 1.

The thematic analysis obtained from the face-to-face interviews categorized the responses into 5 themes, as shown in Figure 1, below. Some of the themes were further categorized into subthemes.

### 3.1. Social Interactions

Social distancing was implemented during the COVID-19 pandemic as a restriction measure to reduce physical contact, thereby reducing the spread of infection. This, however, adversely affected mental wellness and comfort during such difficult times.

#### 3.1.1. Decline in Interactions

All participants mentioned that their social interactions were affected and that this had had an adverse effect on them. However, they adhered to the social restrictions as they did not want to be infected. This resulted in some friendships being dissolved due to the lack of interaction.

“*Our social interaction was affected badly because we could not visit each other the way we were used to, no visitation no checking upon each other, nothing. Some friendships died during that period, but there was nothing that we could have done, each person was protecting him or herself*.” (P9: Age, 79)

Masking had numerous adverse effects, such as not recognizing others who were known to them. It also made hearing the other person difficult.

“*Our social life was also very disturbed, with all the masks we could not even hear each other when we talk, sometimes we could not recognize each other in the passages because of masks*.” (P11: Age, 89)

The participants noticed alterations in children’s behaviour as a result of the lack of socializing. As these were unfavourable changes, eventually, measures were advocated to allow children to socialize. Other precautionary measures were taken to ensure the prevention and spread of disease.

“*I saw changes in my grandchildren, children like to go out and play and do all outside activities, but with the restrictions they could not. I noticed changes in their behaviour; those changes were not good. They started acting reserved and down, there was nothing I could do; I started allowing them to go to the park and play. I ensured that they wore masks, and they would go straight to the bath when they come back and change clothes immediately*.” (P12: Age, 68)

The participants themselves also experienced other difficulties due to constantly remaining indoors. The lack of walking or exercise exacerbated conditions such as arthritis.

“*Remaining indoors was difficult for me. I have arthritis that gets worse if I sit or remain in one position for too long, so I was in constant pain because I could not go out for my regular walks around the neighbourhood. I could not socialise with the neighbours as well*.” (P10: Age, 76)

#### 3.1.2. Caring for Others

During those times when the participants’ assistance was required, social restrictions were not adhered to. They felt that it was particularly important to care for others, particularly their neighbours.

“*Some of the neighbours here stay alone and we are the only people they have to help them. Because of Ubuntu (humanity) we could not just neglect them. With those few visits, we still wore masks and sanitized*.” (P10: Age, 76)

Another participant was the recipient of similar care from her neighbours. Telephonic and social media communication was also used to maintain contact with others, which helped alleviate the loneliness.

“*I had neighbours that were constantly checking up on me and were taking good care of me, I never felt alone. I would still communicate with my family via cell phone calls and WhatsApp*.” (P3: Age, 60)

### 3.2. The Effect of High Death Rates

During the initial spread of the pandemic, there was an exponential increase in incidence rates, which led to high death rates related to COVID-19. National lockdown requirements included restrictions on gatherings, which included funeral attendance. Participants were emotionally conflicted regarding attendance and safety, especially with deaths related to COVID-19.

“*Another thing that was very sad is the restrictions at funerals. It was very difficult because only a certain number of people would go to funerals. During her (cousin’s) funeral, her coffin was wrapped in plastic and the coffin looked as if it was smaller than her. It was heart-breaking, that is what broke my heart the most, the way my cousin was buried. We also could not attend other family funerals outside of KZN because of COVID-19*.” (P6: Age, 81)

One participant lost her offspring to COVID-19. She mentioned the difficulty of planning a funeral for a small group of people.

“*Two of my children got sick and one unfortunately passed away and he was infected with COVID-19. Planning the funeral was difficult and the limited number of people allowed to attend made it worse. As a family alone we are more than 50, it was very difficult to choose who comes to the funeral and who doesn’t*.” (P11: Age, 89)

The inability to attend funerals led to much emotional strain, as alluded to by this participant.

“*The virus was very bad; I would cry for people. We did not know how it came. We never saw my aunt and uncle; we did not go to their funeral. My mother is 90 and she never got to see her sister’s funeral, we all never went there*.” (P2: Age, 55)

### 3.3. Financial Implications

#### 3.3.1. Financial Support

The majority of participants (*n* = 9) in this study were pensioners above the age of 60 years. They continued receiving their pension; thus, their income was secure.

“*It did not affect my income; I still got my pension grant with no issues*.” (P9: Age, 79)

In South Africa, people who have previously been employed receive funds from a government unemployment insurance fund (UIF). This continued during the pandemic, as mentioned by this participant whose husband received UIF.

“*I still got my retirement pension with no issues. My husband was also receiving the temporary UIF funds that they were receiving during the period they could not go to work. We still managed fine financially*.” (P8: Age, 60)

However, this funding may not have been consistent since one participant mentioned that the UIF funds were not received by her family members who lost their jobs during the pandemic.

“*Financially, it was painful because there were some of my family members and neighbours that could not continue working and did not even receive their UIF money, that some people were receiving to support themselves during that period of being unable to work. It was difficult for them*.” (P4: Age, 65)

In vulnerable settings, community members were supportive of one another during the pandemic, offering assistance where possible, especially to those who were affected by the loss of their primary income. Some participants received additional financial support from their families, who, in addition, checked on their well-being.

“*My son was able to support me financially. There were also some people from a Pick ‘n Pay supermarket from Joburg that bought groceries for me, and they sent food regularly. My son made sure that we had everything we needed. They would always phone and check, “Mom, you got this? What did you cook today? What did you eat today?” They were all there for me. I can say they always checked on us, always*.” (P1: Age, 63)

#### 3.3.2. Loss of Additional Income

The women mentioned the risk of unemployment due to the closing of businesses during the COVID-19 lockdown restriction period. This risk was higher for those who were younger than 60 years of age as they were contributing to the country’s economy. Unemployment would result in a complete loss of income. Fortunately, all participants in this age category did not lose their employment.

“*Other people lost their employment; it was a difficult time. My husband’s company was also affected, but fortunately they never lost their jobs*.” (P5: Age, 54)

Although many received the government pension, they were unable to perform other work to supplement their income, which imposed a financial strain as government pensions are often low and are not sufficient to cover all living expenses.

“*The most difficult thing was the inability to continue with the work that we were doing. We have a group of ladies and children that run a sewing project of different clothing garments and sell them every Sunday at the marketplace in town. The sewing project had to stop because of restrictions. We were left with nothing to do. Finances were also affected, we were no longer selling our garments, that meant no money was coming in, we only were dependent on the government pension. Everything needed money—from groceries, medications, rent, electricity, and everything, but there was so little that we had at that time. It was difficult*.” (P12: Age, 68)

Pensioners’ incomes were sometimes supplemented by their family members. This was discontinued as the relatives’ own incomes were reduced due to a loss of employment, resulting in financial strain.

“*I still received my pension. It is my children that lost their employment and business, but still that affected me because they were no longer able to give me money as they used to before*.” (P3: Age, 60)

#### 3.3.3. Additional Expenses

Appropriate hygiene was a major driver of reducing infection rates, according to government health recommendations, but it was considered an additional cost to maintain barriers against the transmission of COVID-19.

“*Financially, it was bad because as they said, we were supposed to keep washing and keep cleaning ourselves and our households, so we needed extra money for all those cleaning things, that was a bit hard*.” (P1: Age, 63)

However, they also felt a financial strain due to increased expenses as they were housebound and, thus, snacking more often than usual. The continuous use of the television and other household appliances also increased the electricity consumption.

“*The only challenge was that we used more money during that time because if you are in the house all day you tend to eat more, get more cravings and you want to snack all the time because there is nothing keeping you busy. We would watch TV all day, use all the home appliances all the time, and that increased the electricity bill and water bill*.” (P8: Age, 60)

### 3.4. Anxiety and Stress

Governmental safety measures such as the wearing of masks led to discomfort, which then led to anxiety.

“*During that time, you couldn’t go out and at times you couldn’t go to do your grocery shopping at any time. We had to wear a mask all the time. Those restrictions we had made me feel very sad and upset, but in another way, it was for our own safety as older people, the older we are, the easier we can get sick, I always thought about that*.” (P1: Age, 63)

Other government restrictions that limited people from moving about also led to stress and anxiety. Older people who may not have been computer literate had difficulties in purchasing required items online. They felt deprived of their freedom.

“*It was difficult to also go to the shops, especially for us old people who do not know how to buy online and get your things delivered. I felt like freedom was very limited, we could not do most of the things that we wanted to do and were used to doing*.” (P9: Age, 79)

News of the infection and or hospitalization of family and friends was stressful, particularly since the participants were unable to visit loved ones who were ill, leading to them feeling very restricted.

“*It was very hard when we heard of all the people in hospitals, it was hard. One minute we would be talking, the next minute you would hear something has happened to someone close to you. It was very hard and hurting because we are a family…. it was also difficult for me to go to hospital. It was like we all are tied up*.” (P2: Age, 55)

### 3.5. Coping Mechanisms

Stress during COVID-19 was alleviated by remote interactions, which included telephonic communication among family members. This provided a means of communication and reassurance that family members were well and safe.

“*Most of the time, my family would phone me. Even my kids would phone me every day. My son, my daughter-in-law, and my daughter would phone me every day and check on us, So, I think those phone calls also helped a lot. I would know they are still in contact and they are still worrying about us*.” (P1: Age, 63)

#### 3.5.1. Self-Care

Many participants used hobbies to keep themselves busy as a means of self-care.

“*I love cooking and baking, so I was constantly on the stove all the time, and that helped me to keep myself busy and have less time to worry*.” (P8: Age, 60)

Another participant reiterated that she used this hobby to supplement her income. She also taught her children and grandchildren the art of sewing.

“*I tried to continue with the sewing project from my house, but it was very difficult without the other ladies. I just needed something to do just to pass time. I then started sewing masks after seeing that they were in demand as everyone had to wear them. The masks sewing project ran successfully for a whole year, we had started to make money, until unfortunately I started developing arthritis that made it very difficult for me to continue. My coping mechanism was continuing with my sewing project and teaching my kids and grandkids to sew as well*.” (P12: Age, 68)

#### 3.5.2. Religious Belief

Religious belief was identified as a coping mechanism to deal with the stress associated with the COVID-19 pandemic. This was largely due to concerns about the safety of oneself as well as that of others at risk of contracting the virus.

“*I just prayed. I prayed so much; I prayed every day. So, you would say prayer was my coping mechanism, it helped me a lot*.” (P1: Age, 63)

“*I am a born-again Christian, if it was not for God I don’t know if I would have survived. What we could not handle we just told God, we needed God to give us strength and give us wisdom and tolerance for each other*.” (P6: Age, 81)

#### 3.5.3. Treatment

Home remedies were commonly used by all the participants to prevent an infection and as a curative mechanism by those who were infected.

“*We regularly used some of the home remedies that have been known to prevent and treat flu, for example, ginger, garlic, lemon and honey. I never got sick with any other illnesses during that time*.” (P11: Age, 89)

Others realized the importance of boosting their immune system by taking multivitamins.

“*I also took multivitamin tablets to keep my immune system strong enough*.” (P8: Age, 60)

## 4. Discussion

This study explored the wellness and lived experiences of vulnerable women in communities that have been affected by COVID-19 in South Africa. The findings highlight the psycho-social impact of the pandemic on this community.

In this study, we report the experiencing of mental health challenges, including stress, and anxiety, due to the initial lockdown procedures that were implemented. The strict ‘stay at home orders’ isolated people from their loved ones and prevented social interactions, thereby disrupting many relationships, particularly among friends. The use of masks created barriers to social connectedness and prevented effective interpersonal communication. Such a disruption of relationships has been reported to cause stress and depression, particularly among women [10,30]. Social distancing and other safety measures have also been shown to negatively affect psychological and social well-being [31]. However, we show here that phone calls played a crucial role in helping people stay connected during the COVID-19 pandemic, especially when social distancing and lockdown measures were in place. Phone calls provided a way for people to maintain their social connections and combat their feelings of isolation. Engaging in conversation with loved ones helped alleviate the emotional toll of being physically distanced from them, particularly when people called regularly to check on the well-being of their family. These enquiries about health provided emotional support during a time of uncertainty.

Although the phone calls and text messaging helped to maintain contact, an unfavourable consequence was news of either the hospitalization or death of loved ones. Other studies have shown that the more closely people followed the COVID-19 news, the higher the level of anxiety, which was elevated in those who had at least one relative or friend infected with COVID-19. The severity of anxiety was also greater in high-prevalence areas [30]. Hence, in the year 2020, the World Health Organization (WHO) urged the general population to reduce their search for continuous updates on COVID-19 as it increased depression and anxiety among individuals [12]. Fear related to the possibility of infection has been associated with many mental health challenges. Nxumalo et al. [32] also revealed the presence of stress, depression, and anxiety associated with the death of loved ones. This concurs with previous reports of additional stress being associated with the inability to be with loved ones who were severely ill or dying, necessitating psychosocial interventions to address this emotional trauma and support family members [31].

Studies have also shown that COVID-19-related unemployment increased the negative effects on mental health [2,8,21]. However, this was not identified as a major contributor to stress in the present study, as the majority of participants were the recipients of pension grants. Historically, women have experienced various gender inequalities throughout their lives, showing significant constraints in their older age. However, governmental social pensions are designed to not discriminate, allowing women to have access to basic financial security in their old age [17], Social grants decrease the incidence of depression, especially among those from low-income households, as reported in a previous South African study [8]. Nevertheless, the overall income was reduced as most participants supplemented their cash inflow with some form of supplementary work, which had to be discontinued during stringent lockdown measures. Those who received financial support from their children were also negatively impacted as this supplementary cash inflow was either decreased or stopped as younger family members lost their jobs. This was a worldwide phenomenon, as shown by an American study, which revealed that 33% of the population had either lost their jobs or had taken a pay cut [33].

This study’s findings indicate that financial strain was exacerbated by increased expenditure during the pandemic, particularly on non-essential items. This was also noted in other studies, which showed unusual spending patterns due to health-related expenditure and new care-related responsibilities [34,35,36].

Among our study participants, mental health was improved by establishing new routines involving hobbies such as baking and sewing. It is notable that the latter was used to supplement income by sewing cloth masks. Teaching these skills to younger family members created a sense of intimacy. The new routines also included virtual methods of maintaining friendships, with increased telephonic or social media use, which, to some extent, alleviated the loneliness. The 2020 WHO report also advocated supporting others via telephonic conversations among neighbours and community members to improve the indicators of wellness. The report also conveyed messages encouraging those who were in isolation to stay connected to their social networks while limiting physical interaction [1].

Within this investigation, religious belief was identified as a coping mechanism to combat stress. People turned to God in prayer to seek ease from the worries of the pandemic. One participant, in particular, mentioned that her Christian faith helped her to cope. Other studies have also shown that those with strong religious beliefs were able to cope better. For instance, communities that followed the Islamic faith showed a higher ability to cope with stressors relating to COVID-19 [37,38]. Ribeiro et al. [39] further indicated that religion and faith can counter the negative effects of various stressors.

The focus on maintaining good health and enhancing the immune system is evident, as numerous participants opted for multivitamins in an attempt to strengthen their immune systems. However, some participants used non-pharmacological treatments including ginger, garlic, and honey for the alleviation of potential and perceived COVID-19 symptoms. Both ginger and garlic are believed to have potential health benefits as they contain various bioactive compounds that have anti-inflammatory and anti-microbial properties [40,41]. However, their efficacy in boosting the immune system remains to be investigated.

The study was not without its limitations. For this research, a qualitative study was conducted among a vulnerable group in one setting only. The researchers were cautious and showed awareness when addressing certain topics of discussion that may affect participants’ well-being, such as their wellness and afflictions during COVID-19. The interviews were conducted sympathetically to reduce the impact of any sensitive situations that occurred by providing reassurance and comfort where needed by the participants. All interactions were documented and cross-analysed with the data to ensure that the participants’ true experiences were captured. Although the sample size was small, data saturation was attained in this group of individuals, which made the thematic reporting uniform.

## 5. Conclusions

We conclude that among aging vulnerable populations, isolation from loved ones and news of deaths were major stressors during the COVID-19 pandemic. When considering the affliction of COVID-19 among older women in vulnerable settings, the identification of various stressors may present mental health challenges and coping difficulties post-pandemic. Ideally, these findings should inform policy development and implementation to improve access to care facilities, especially for older people from vulnerable communities. During the COVID-19 pandemic, emergency preparedness for supporting elderly groups did not provide clarity and exclusivity for the older population, apart from their being a group that was highly vulnerable to infection. Mental health and care should be a point of focus in future emergency preparedness.

## Figures and Tables

**Figure 1 ijerph-20-07040-f001:**
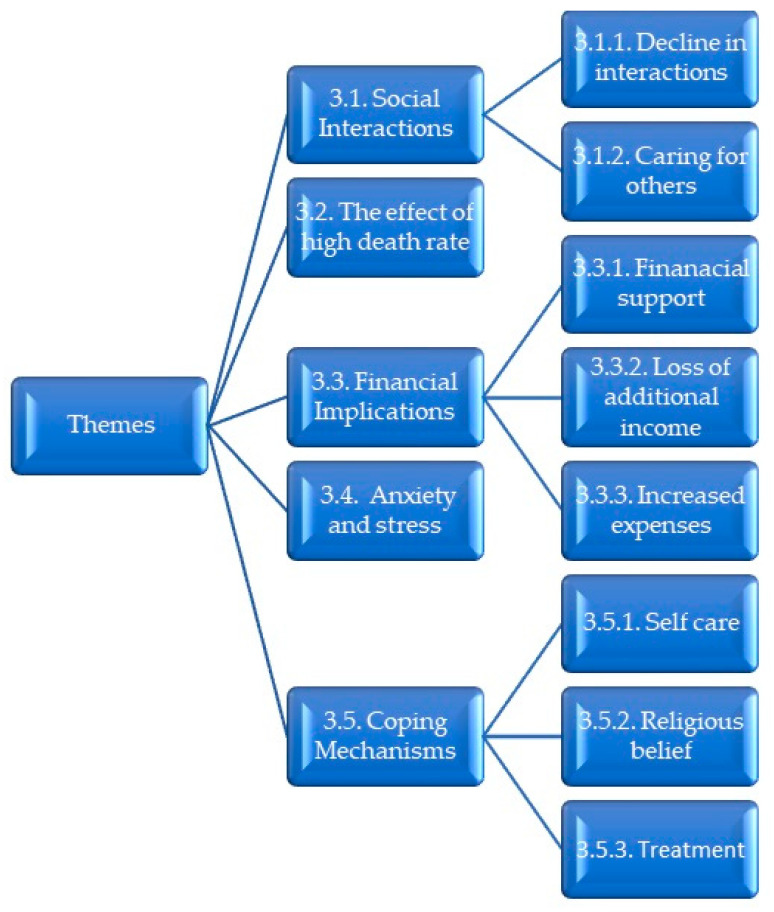
Flow diagram, indicating the themes and sub-themes.

**Table 1 ijerph-20-07040-t001:** Demographic characteristics of participants (*n* = 12).

Research Participant	Age	Ethnicity	Relationship Status	Number of Children	Persons in Household	Source of Income
P01	63	Indian	Widowed	2	2	Government pension
P02	55	Indian	Widowed	1	5	Government pension
P03	55	Black	Single	7	5	Government pension
P04	65	Black	Single	2	2	Government pension
P05	54	Black	Married	2	4	Disability grant
P06	81	Caucasian	Married	7	6	Government pension
P07	74	Black	Cohabiting	1	3	Government pension
P08	60	Black	Married	3	5	Retirement pension
P09	79	Black	Single	1	2	Government pension
P10	76	Black	Divorced	None	4	Government pension
P11	89	Black	Single	None	3	Government pension
P12	68	Black	Married	6	6	Government pension

## Data Availability

The data presented in this study are available upon request from the corresponding author. The data are not publicly available due to privacy restrictions.

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
