# Peer review of "Exploring the Lived Experiences of Vulnerable Females from a Low-Resource Setting during the COVID-19 Pandemic"

_ijerph, 2023, doi:10.3390/ijerph20227040_

Round 1

Reviewer 1 Report

Comments and Suggestions for Authors

The manuscript “Exploring the lived experiences of elderly females from a low resource setting during the COVID-19 pandemic” is significant in the context of emergencies among vulnerable populations. However, the authors need to address the following points:

1.     Mention about SRQR or COREQ

2.     Data management and analysis section need to improve.

3.     The results section there is a lot of quotations, the authors need to reduce some quotes and the results section needs more descriptions.

4.      In the Discussion Methodological consideration section is missing

5.     What is the implication of findings for future emergency preparedness?

6.     The conclusion section is only one sentence. 

Comments on the Quality of English Language

Minor editing of English language required

Author Response

Responses to Reviewer 1 Comments

Point 1: Mention about SRQR or COREQ.

Response 1: Thank you for the suggestion. The study adopted the SRQR developed by O’Brien et. al. (2014). We have stated this in lines 139-140.

Point 2: Data management and analysis section need to improve.

Response 2: Thank you. Improvements have been made in the data collection section for data management and analysis. See lines 129-132 and 137-139.

Point 3: The results section there is a lot of quotations, the authors need to reduce some quotes and the results section needs more descriptions.

Response 3: Some of the quotes have been shortened and some removed. However, we cannot reduce these any further as it is important to substantiate results with quotes in qualitative research.

Point 4: In the discussion, methodological consideration section is missing.

Response 4: Yes. Thank you. These are included in the last paragraph of the discussion.

Point 5: What is the implication of findings for future emergency preparedness?

Response 5: This has been addressed as recommendations in the conclusion.

Point 6: The conclusion section is only one sentence. 

Response 6: Refer to point 5.

Reviewer 2 Report

Comments and Suggestions for Authors

There are some comments the authors must respond and major revision is necessary!

Introduction

The introduction should be more structured according to 5 main themes explored in the paper (social interactions, high death rate, financial implications, mental health, and coping mechanisms). So, more references are needed about the mentioned themes in South Africa and precisely about the target group.

·       Put the aim of the study at the end of the introduction section and state it clearly! The aim of the paper should be more specific and more connected with the title of the paper! The study was conducted in a private room at the Homeopathy clinic, a low socio-economic housing estate in Durban so the patients from poor settings and affected by COVID-19 are the main target group.

·       Please, explain what is KZN province! Upon the first appearance in the text, you should explain that abbreviation. And what is the prevalence of COVID-19 in that province in figures?

·       What kind of COVID-19 measures (governmental safety measures) were implemented during the July-August 2023 when the study was conducted? The authors mentioned some at the beginning of the discussion section and they should be explained in the introduction!

·       Please, define clearly the vulnerable population group/studied vulnerabile community! In what sense they are vulnerable? This could be the double or triple burden of vulnerability like they are women, from poor settings and older women with probably some CNDs and also homeopathic patients. If these are elderly they should be 60 or 65 age and more!  In the paper, they are aged 50 years and over.

Method

·       One or two paragraphs are needed about variables and questions regarding 5 main themes explored in the study!

Discussion

·       More specific references are needed!

·       The authors should begin each paragraph with the results from their study (or when this is the case like at the beginning of the 4 and 5 paragraph they should add "in this or our study") and then compare the obtained results with other references!

·       Conclusion is too general and debatable concerning a small sample, only 12 interviewed women! "We conclude that among elderly vulnerable populations, isolation from loved ones and news of death were major stressors during the COVID-19 pandemic".

One paragraph regarding recommendations and policy implications is lacking!

Comments on the Quality of English Language

Minor editing of English language required

Author Response

Responses to Reviewer 2 Comments

Point 1: Introduction:  The introduction should be more structured according to 5 main themes explored in the paper (social interactions, high death rate, financial implications, mental health, and coping mechanisms). So, more references are needed about the mentioned themes in South Africa and precisely about the target group.

Response 1: Thank you for the suggestion. Literature has been improved to include various themes explored.

Point 2: Introduction: Put the aim of the study at the end of the introduction section and state it clearly! The aim of the paper should be more specific and more connected with the title of the paper! The study was conducted in a private room at the Homeopathy clinic, a low socio-economic housing estate in Durban so the patients from poor settings and affected by COVID-19 are the main target group.

Response 2: Yes, Thank you. Correction has been made.

Point 3: Introduction: Please, explain what is KZN province! Upon the first appearance in the text, you should explain that abbreviation. And what is the prevalence of COVID-19 in that province in figures?

Response 3: Yes. Thank you. Correction has been made. KZN given in full and the prevalence stated in line 82.

Point 4: Introduction: What kind of COVID-19 measures (governmental safety measures) were implemented during the July-August 2023 when the study was conducted? The authors mentioned some at the beginning of the discussion section and they should be explained in the introduction!

Response 4: The study was conducted in July-August 2023, after the government restrictions were lifted. However, we sought to assess participants’ experiences during COVID-19 lockdown from the period of March 2020 onwards to March 2022. (See lines 80-82). Interviews were conducted post governmental COVID-19 restrictions during July-August 2023. (See lines 92-94)

Point 5: Introduction: Please, define clearly the vulnerable population group/studied vulnerabile community! In what sense they are vulnerable? This could be the double or triple burden of vulnerability like they are women, from poor settings and older women with probably some CNDs and also homeopathic patients. If these are elderly, they should be 60 or 65 age and more!  In the paper, they are aged 50 years and over.

Response 5: Yes Thank you for the comment. Vulnerable social groups include older women aged 50 years and over that were from low-income settings. (Added to aim). Paragraph included in introduction focusing on vulnerable groups (Line 57- 77). Further outline in methods

Point 6: Methods: One or two paragraphs are needed about variables and questions regarding   main themes explored in the study!

Response 6:  Thank you for the suggestion. This has been included in the data collection subsection. (line 128-134)

Point 7:Discussion: More specific references are needed!

Response 7: Thank you. Additional references have been included.

Point 8:Discussion: The authors should begin each paragraph with the results from their study (or when this is the case like at the beginning of the 4 and 5 paragraph, they should add "in this or our study") and then compare the obtained results with other references!

Response 8:Yes. Thank you for the suggestion. This has been ammended in the discussion. Additional references have been included to outline multiple study comparisions.

Point 9:Conclusion: Conclusion is too general and debatable concerning a small sample, only 12 interviewed women! "We conclude that among elderly vulnerable populations, isolation from loved ones and news of death were major stressors during the COVID-19 pandemic".

Response 9: The conclusion has been expanded based on the findings of the study. Recommendations have been addressed in conclusion.

Point 10:Conclusion: One paragraph regarding recommendations and policy implications is lacking!

Response 10: Yes, Thank you for the suggestion. Recommendations, particularly related to emergency preparedness policies have been expressed.

Reviewer 3 Report

Comments and Suggestions for Authors

In the manuscript, the authors reported a qualitative study on mental health and well-being among older women in the context of the COVID-19 pandemic. The manuscript was well written and explore an pressing issue in gerontology and health care. Using the qualitative method, the study added to literature of life stress and mental health in the aging population and has practical implications for coping with the lasting effect of the pandemic. However, due to its qualitative nature, it is difficult to draw the strong conclusion at the end of the article.

1. In the Introduction, it'd be helpful to switch the focus from COVID-19 to older adults. Because this population is extremely vulnerable not only in the pandemic but also in all kinds of public health crisis. The focus on older population would be easier to argue for the external validity of the study.

2. The further justification for the use of the sample collected from the local community is needed. The sample may entail certain features in a specific country or region, which may prevent the generalization of the findings.

3. Please elaborate on the snowball sampling used in the present study.

4. Were there any standardized scores of life stress, anxiety or depression?

5. It's expected to interpret the results with existing quantitative studies in the discussion.  

Author Response

Responses to Reviewer 3 Comments

Point 1: Introduction:  In the Introduction, it'd be helpful to switch the focus from COVID-19 to older adults. Because this population is extremely vulnerable not only in the pandemic but also in all kinds of public health crisis. The focus on older population would be easier to argue for the external validity of the study.

Response 1: Thank you for the suggestion. Literature has been directed towards elder women. See line 66-74.

Point 2: Methods: The further justification for the use of the sample collected from the local community is needed. The sample may entail certain features in a specific country or region, which may prevent the generalization of the findings.

Response 2: Thank you.Justification provided. See lines 99-103.

Point 3: Methods: Please elaborate on the snowball sampling used in the present study.

Response 3: Additional explanation of the process has been discussed in the methods.

Point 4: Methods: Were there any standardized scores of life stress, anxiety or depression?

Response 4: No, due to the study design, we were unable to implement a depression, anxiety and stress scale to measure outcomes based on the statistical output for qualitative data, rather we explored narrative experiences.

Point 5: Discussion: It's expected to interpret the results with existing quantitative studies in the discussion.

Response 5: The discussion has been improved by interpreting the results in more detail. Existing studies have been cited.

Reviewer 4 Report

Comments and Suggestions for Authors

Gerontology assumes that old age begins at the age of 60 or 65. In the studies, as many as 25% (3) of people were under 60 years of age (55, 55, 54).

The discussion is too general and requires careful analysis. Also the Conclusions part is too general and short.

The study group consists of only 12 people. Although these are qualitative studies, in my opinion, the article does not contribute much to science and is not suitable for publication in this form.

Author Response

Responses to Reviewer 4 Comments

Point 1: Introduction: Gerontology assumes that old age begins at the age of 60 or 65. In the studies, as many as 25% (3) of people were under 60 years of age (55, 55, 54).

Response 1: Thank you for the suggestion. This has been ammended to discussed vulnerable women (includes change in title of study).

Point 2: The discussion is too general and requires careful analysis. Also, the Conclusions part is too general and short.

Response 2: Thank you. Improvements have been made. Recommendations have been included in conclusion.

Point 3: The study group consists of only 12 people. Although these are qualitative studies, in my opinion, the article does not contribute much to science and is not suitable for publication in this form.

Response 3: Qualitative studies provide information about the lived experiences of participants. In qualitative studies, the sample size can be low as long as data saturation is achieved. Data saturation was achieve in this study.

Round 2

Reviewer 1 Report

Comments and Suggestions for Authors

Thanks for addressing all the suggestions.

Comments on the Quality of English Language

  Minor English editing is needed. 

Reviewer 2 Report

Comments and Suggestions for Authors

I wish to thank the authors for the improvements made! Congrats!

Reviewer 3 Report

Comments and Suggestions for Authors

The issues have been adequately addressed. The study will add to the current literature of psychosocial factors of health outcomes of COVID-19.

Reviewer 4 Report

Comments and Suggestions for Authors

After renaming the article, I believe that this version can be published in the Journal. The authors have made a correction and the article may now be of interest to readers.